# A scoping review of Deaf awareness programs in Health professional education

**Julia Terry**[1]*, **Rhian Meara**[2]

**1** Faculty of Medicine Health and Life Science, Swansea University, Swansea, Wales, United Kingdom,
**2** Faculty of Science and Engineering, Swansea University, Swansea, Wales, United Kingdom

\* j.terry@swansea.ac.uk

**Data Availability Statement:** Submitted as Supporting Information files.

**Funding:** This work was supported by the Burdett Trust for Nursing, grant number 101010662 \737073 – JT. The funders had no role in study

## Abstract

Deaf awareness aims to promote understanding about Deaf and hard of hearing people, with the goal of reducing barriers between Deaf and hearing populations; and is particularly pertinent for health professional students as they need to learn to communicate effectively with a range of population groups. This scoping review aims to provide an overview of literature examining Deaf awareness programs provided to health professional students during their initial training. We searched four medical and public health databases and registers using terms related to Deaf awareness. We used the PRISMA-ScR reporting standards checklist for scoping reviews. We identified 10,198 citations, with 15 studies included in the final review. Searches were performed during August to September 2022, and April 2023. Studies were included provided they examined Deaf awareness content or programs within health professional education. Data were extracted by two independent reviewers who screened all abstracts using Rayyan software, followed by discussion to achieve knowledge synthesis and agreement. In all, a total of 15 articles from six countries were identified across health professional student disciplines including pharmacy, nursing, audiology, interprofessional and medical programs. The review found sparse evidence of research into Deaf awareness programs delivered to health professional students, with delivery often solely to small groups of students, indicating why so few students can access information about how to communicate effectively with Deaf and hard of hearing patients during their initial training programs. This scoping reviewed showed evidence of promising benefits for health professional students undertaking Deaf awareness programs during their undergraduate education. The importance of communicating with Deaf and hard of hearing patients and attaining Deaf cultural competencies for health professional students should be investigated in future research.

## Introduction

Deaf awareness training aims to promote understanding about Deaf and hard of hearing people, with the goal of reducing barriers and increasing accessibility between Deaf and hearing populations and combating discrimination. In this paper the terms Deaf and hard of hearing

design, data collection and analysis, decision to publish, or preparation of the manuscript.

**Competing interests:** The authors have declared that no competing interests exist.

people will be used throughout. The heterogeneity of Deaf and hard of hearing people are often not fully known to health staff [1] and that a person-centred approach is required according to individual communication needs. People who identify as culturally and pro-foundly Deaf may be referred to using a capital D for Deaf, with a lower case 'd' for deaf, more commonly used for people who are hard of hearing.

Despite the numbers of Deaf and hard of hearing people increasing globally from around 466 million people currently, to 900 million by 2050 [2], there is limited preparation and train-ing for health professionals and health professional students to communicate and work with Deaf patients [3–5]. Health professionals themselves have reported that their communication skills and knowledge of working with Deaf and hard of hearing people could be greatly improved [6], as they lack tailored communication skills in caring for Deaf and hard of hearing populations [7]. There is a dearth of evidence about the effectiveness of provider-oriented dis-ability programs, specifically those relating to sensory loss, and those in existence tend to be focused on general disability awareness [8] or about attitudes and behaviours towards specific population groups [9,10].

Challenges for Deaf and hard of hearing patients are many and are often not known to health professional groups with whom they engage. Difficulties for Deaf and hard of hearing people often relate to a lack of accessible services and provision in education [11], in society [12], and in disaster response [13] as well as difficulties accessing health services [14,15]. It is also acknowledged that health professional students need to be trained in Deaf cultural compe-tencies [1] so they develop relevant knowledge and skills about Deaf and hard of hearing cul-ture. For example, a person may use a Signed language as their preferred communication method, whilst others may prefer information literature in written form. However, literacy lev-els in Deaf and hard of hearing people are often lower than in hearing populations [16], so it is essential that health workers learn to ask about preferred communication methods for each individual. Students may demonstrate attitudes to Deaf people stemming from their own lack of knowledge that results in a negative stigma toward anyone who is Deaf (referred to as audism) [17], particularly if they lack experience of working with Deaf and hard of hearing patients [18].

Further challenges are reported by Deaf and hard of hearing patients who note they do not understand health providers instructions in nearly half of appointments, with few clinicians checking patient understanding [19] suggesting potential risk of misunderstanding and resul-tant health risks. Difficulties often result when a Sign language interpreter is required as health staff have little notion how to book or how to work with a Deaf patient and a Sign language interpreter [20]. Similarly, few health professional staff have used remote video interpreting services during health consultations [21], which involves either the health facility or the patient using a sign language interpreter via an app or remote video interpreting service (either in a booked capacity or on-demand). Few health professionals or students know the challenges members of Deaf and hard of hearing communities experience accessing health services, and specifically care routes that may or may not be open to them [22].

Notably in healthcare settings few staff have Deaf awareness training which leads to persis-tent health inequalities for Deaf and hard of hearing patients who often have poor experiences and outcomes in healthcare settings [23]. These negative experiences can relate to discrimina-tion around booking procedures and face to face appointments, as well as assessments and testing visits [24,25], often due to limited accessibility for communication options [26], with services unprepared and ill-equipped to meet the needs of Deaf and hard of hearing people [27]. It is imperative that health service experiences improve for this population group. It has been described as a silent epidemic with global efforts needed to address the unmet needs of Deaf and hard of hearing adults and children who experience poorer health and care [28].

The Deaf awareness knowledge gap is likely unknown by health workers, who may have had limited exposure to this population group, and consequently do not appreciate the healthcare barriers Deaf and hard of hearing people experience [29]. Furthermore, there is increasing evidence that Deaf and hard of hearing people experience poorer health, with increased risk of preventable ill-health with chronic illness often undiagnosed and untreated, such as diabetes and cardiac disease [22]. Many diverse groups are disadvantaged because of assumptions around health literacy that may relate to English not being a first language, and ability to read and write, which in turn impacts on a person's ability to understand healthcare and pharmacy directions and information [30]. Deaf people are aware health information is often not in accessible formats, so consequently they may rarely seek health material and be disadvantaged as a result by not being aware of common risks or solutions within their own control.

Knowledge of Signed language and the use of telecommunication equipment, such as Sign language relay services is not prevalent in health providers [31]. Deaf awareness programs highlight the different forms of communication that Deaf and hard of hearing people may use [32], including sign language, lip reading, note taking and oral methods, but few health workers are aware of this. Individual education providers may offer opportunities for students that challenge their knowledge about diversity, increase knowledge and communication, and break down stereotypes [33]. Certainly, there is a need for increased disability training in health professional education [34], with the most effective programs noted to be those that include people with disabilities themselves.

The aim of this scoping review was to report on the published evidence of Deaf awareness programs experienced by health professional students during their initial training. Given the health inequalities that Deaf and hard of hearing people experience, we wanted to explore the range of interventions and approaches used with health professional students to understand the current evidence about Deaf awareness programs.

## Methods and analysis

### Ethics statement

As this study only included published data, ethics approval was not sought. The methods and results are reported according to the relevant items of the Preferred Reporting Items for Systematic reviews and Meta-Analyses extension for Scoping Reviews (PRISMA-ScR) checklist [35]. According to Verdejo et al. [36] the main aim of a scoping review is to identify and map the available evidence for a specific topic area. The approach to the review was based on Arksey and O'Malley's framework [37] which consists of the following stages: i) identifying the research question; ii) identifying relevant studies; iii) selecting studies; iv) charting the data; and v) collating, summarising, and reporting the results.

### Search strategy

A scoping review seeks to present an overview of a potentially large and diverse body of literature pertaining to a broad topic, whereas a systematic review attempts to collate empirical evidence from a relatively smaller number of studies [38]. This scoping review is not intended as a conclusive synthesis of evidence but does provide an overview of the evidence of Deaf awareness programs that exist, primarily for health professional students. The study has been funded by the Burdett Trust for Nursing and was conducted in Wales, UK. It was not registered online. The overall project had a steering group which included lay members, Deaf and hearing professionals. The focus of the steering group was on the empirical aspects of our study and building a Deaf awareness course for Wales, UK, with this scoping review discussed at early meetings, and members contributing ideas for search terms.

**Identifying the research question.** The core aim of this scoping review was to find out 'what is the existing evidence on Deaf awareness programs that are included in health professional education training?'. Deaf and hard of hearing people's experiences in health services and poor health literacy are frequently linked to the poor knowledge of health professionals about how to communicate with Deaf and hard of hearing people; including a lack of training for medical and nursing students, and students studying to become allied health professionals [1,39,40].

**Identifying relevant studies.** The scoping review research question was left intentionally broad. The evidence was searched using four electronic databases (CINAHL, MEDLINE, ASSIA and Proquest Central), registers and key journals and repositories (such as PROSPERO), and contact made with key authors; as well as internet site searches for policies and reports. An experienced information specialist's help was sought in reviewing the search strategy tool (PICO framework), which included students (P- population), Deaf awareness (I- intervention), health professional education (C–context); and learning (O–outcome). Search terms used included: Deaf OR hard of hearing or DHH or sensory loss; combining "deaf aware*" OR "deaf culture*") AND ("learn*" OR "educat*" OR "train*" OR "course*" OR "program*" OR "teach*". The databases included were CINAHL, Medline, ASSIA and Proquest Central, as well as Cochrane registers, with searches conducted between August and September 2022; and again in April 2023 (an example of the search strategy for one database is provided as an additional file). Different techniques and terms were used to expand and narrow searches, including tools such as medical subject headings (MESH), Boolean operators and Truncation. Single and combined search terms included key subject areas on: Deaf, hard of hearing, and Deaf awareness. Education related search terms included learning, education, training, course, program and teaching. Limitations were set to include papers in the English Language and research since 2000. Initial searches found papers in languages other than English did not relate to Deaf awareness programs but to Deaf students. Papers not in English language were excluded to reduce volume, and this remains a common decision for researchers [41]. In addition, key journals, professional organisation websites and reference lists of key studies were searched to identify further relevant documents. The final search strategy and terms were agreed and verified by a health librarian.

Inclusion criteria were: published research articles specific to: a) a focus on Deaf awareness, training on Deaf awareness/Deaf culture and b) were published in the English Language between 2000–2023. Exclusion criteria were: papers published before 2000, not in English language, papers without a focus on Deaf awareness, training/courses/understanding Deaf and hard of hearing patient experience for health professional students.

**Study selection.** The initial search produced a total of 10,159 from database searches and 39 from registers. Once duplicates were removed (n = 5804), a further 4049 records were excluded that did not meet the inclusion criteria, 345 publications remained, and titles and abstracts were screened. All 345 records were screened by two separate reviewers independently using Rayyan software [42] and annotated spreadsheets of retrieved papers. We began by excluding sources that did not describe empirical studies of Deaf awareness courses for health professional students, such as opinion articles, newspaper reports, and papers without a Deaf awareness focus. Inter-rater discrepancies were resolved by discussion. 26 records were then removed in line with the eligibility criteria, and the remaining 15 publications are included in this review (see Fig 1).

**Charting the data.** A data-charting form was developed by one reviewer, and then updated iteratively through discussion with a second reviewer. The 15 included sources were charted initially to examine authors, year of publication and country of origin, study design, sample population, study aim and main findings, which was piloted and found to be effective. Through this process sources were all identified as primary research studies. Papers related to

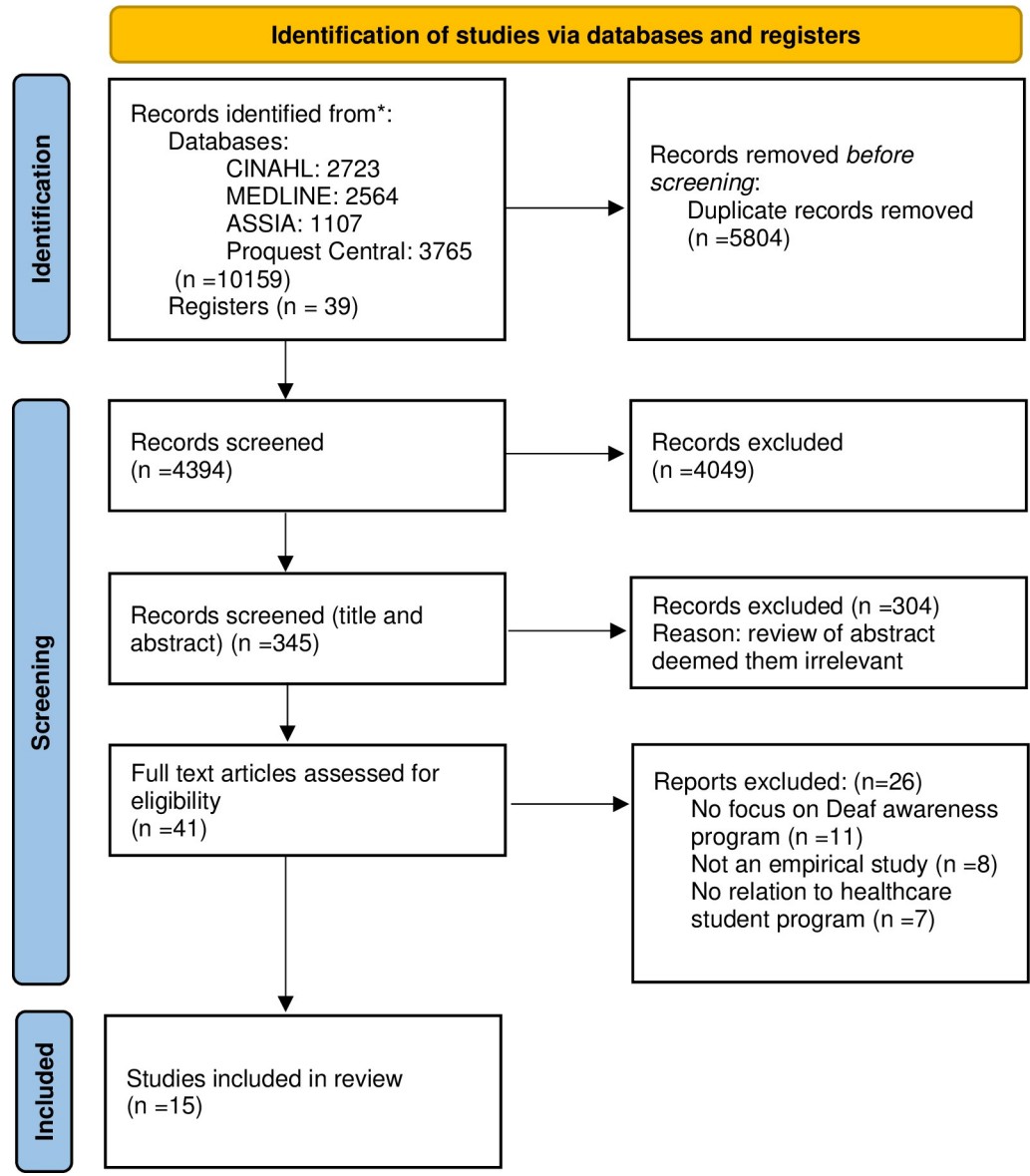

**Fig 1. PRISMA flow diagram: Deaf awareness in health professional student programs.**

the following health professional student disciplines: Pharmacy (n = 2), Nursing (n = 2), Audiology (n = 2), Inter-professional (n = 1), and Medicine (n = 8).

**Collating, summarising and reporting results.** In the final scoping review, six individual countries were represented. Most publications came out of the USA, which may be due to greater funding or interest in this area of research compared to other nations. Due to the heterogeneity of the range of study contexts, a narrative synthesis was a reasonable way to approach the reporting of retrieved studies which included: four pre and post intervention surveys; eight cross-sectional studies; two comparative studies and one evaluation of experiential role play.

After summarising the information from sources, then studies were sorted into categories regarding Deaf and hard of hearing awareness courses for specific health professional education program by discipline, as follows: i) *pharmacy students; ii) nursing students; iii) audiology students; iv) inter-professional students and v) medical students;* and also regarding

methodological approach. In addition, main findings of the sources are presented in Table 1. Context from the grey literature is included in this paper's introduction as this clinical wisdom provides additional information and context.

## Findings

### Identification of studies

The 15 papers included in this scoping review were carried out geographically in the USA (n = 8), Canada (n = 2), the UK including Ireland (n = 3), with one study each from Germany and Puerto Rico. All studies' samples were university students undertaking undergraduate study, and included pharmacy students (n = 3), nursing students (n = 2), medical students (n = 7), and other/mixed groups (n = 5), accounting for some overlap of participant groups.

**Table 1. Deaf awareness studies included in scoping review if change on here, also change in main table, am keeping table for now, just removed figs.**

| Study, year, location | Study design | Sample population | Study aim | Main findings |
|---|---|---|---|---|
| 1. Bailey, N., Kaarto, P., Burkey, J., Bright, D., & Sohn, M. (2021). Evaluation of an American Sign Language co-curricular training for pharmacy students. Currents in Pharmacy Teaching and Learning, 13(1), 68–72.<br>USA | Pre and post educational course survey with identical questions | First and second year pharmacy students (n = 39) | To implement and assess a co-curricular course for student pharmacists to become more confident in communicating with d/Deaf and HOH patients by attending four 90-minute sessions focusing on basic communication and cultural competence | A total of 36 students completed the survey prior to the course, and 34 students completed the survey after completing the course. Pharmacy students perceived an increase in confidence in working with d/Deaf and HOH communities. Authors note that students who signed up for the course were inherently motivated and may have affected the response rate. |
| 2. Diaz, S., & Goyal, D. (2021). Caring for the Deaf: Nursing Students' Knowledge and Awareness. Nursing Education Perspectives, 42(4), 241–242.<br>USA | A 34 item Knowledge of Deaf cultural competency questionnaire | 131 nursing students recruited from one public university in California | To examine Deaf cultural knowledge and awareness in nursing students | Findings showed low knowledge of cochlear implants, use of interpreters and new-born hearing screening rights. There is a need to integrate care for deaf people into all levels of nursing education to promote patient literacy and positive health outcomes. |
| 3. Gilmore, M., Sturgeon, A., Thomson, C., Bell, D., Ryan, S., Bailey, J.,. . . & Woodside, J. V. (2019). Changing medical students' attitudes to and knowledge of deafness: a mixed methods study. BMC Medical Education, 19(1), 1–7.<br>UK | Survey questionnaire to measure attitudes to and knowledge of deafness in those taking an optional deaf awareness course; and focus groups with students to explore ways to incorporate deaf awareness into undergraduate medical curriculum | 64 medical students invited to participate: half on sign language and communication module and the others on alternative module as control. Also students who previously completed the module were contacted to complete questionnaire | To evaluate the impact of specific training on attitudes to and knowledge of deafness, and utilising sign language and communication; and to explore whether a change of attitudes and knowledge persist in the long-term | A significant difference was noted between knowledge scores of those students who were taking the Sign language course and agreed to take part (n = 29) and control group. Focus group data indicated students without knowledge of deafness were uncomfortable communicating with deaf patients and could perceive patient mannerisms as rude. Students reported that without encountering deaf people it may be difficult to understand the issues they face |

(Continued)

**Table 1.** (Continued)

| Study, year, location | Study design | Sample population | Study aim | Main findings |
|---|---|---|---|---|
| 4. Grady, M. S., Younce, A. B., Farmer, J., Rudd, A. B., & Buckner, E. B. (2018). Enhancing communication with the deaf through simulation. *Nurse Educator*, *43*(3), 121–122. USA | Nursing students were exposed to Deaf standardised patients and undertook a history taking exercise, without knowing the patient would be Deaf. After a lecture on communicating with Deaf patients, students undertook the exercise a second time | Nursing students in one USA university (number not stated) | To develop a simulation for nursing students to learn how to communicate with Deaf people | In the initial interaction some students walked out of the room to get an interpreter without saying to the patient they were going for help, and others turned away lessening the chance for eye contact and lip reading. Following the lecture students demonstrated more deaf awareness skills, pointed to their name badge and were better prepared. Authors suggest the simulation could be used for multiple health professions. Benefits were reported by Deaf people involved in terms of improving care for others and having a voice in educating future nurses. |
| 5. Greene, S. J., & Scott, J. A. (2021). Promoting cultural awareness, professionalism, and communication skills in medicine through anatomy: The Deaf culture session. *Clinical Anatomy*, *34*(6), 899–909. USA | Pre and post assessment survey questionnaire | Y1 students (n = 100) Deaf awareness face to face, Y2 students (n = 99) via zoom | To determine the level of pre-existing knowledge of students about deaf people and to evaluate if and what students learned from the session, and to collect feedback | Students rated the session as 4.8 (mean 4.7). 100% Y1 students and 95% Y2 students agreed with the statement to hold the session in the future. Sessions of deaf awareness have the potential to break down barriers that may impact future patient care. |
| 6. Ham J, Towle A, Shyng G. Deaf and hard of hearing awareness training: A mentor-led workshop. The Clinical Teacher. 2021 Apr;18 (2):180–5. Canada | Reflections by students following Deaf awareness workshop | Students (n = 49) from ten different disciplines, including nursing, dentistry, occupational therapy, medicine and social work, attended three pilot workshops | To explore how to develop a Deaf and hard of hearing training workshop, led by Deaf people | Working with a Deaf charity organisation supports delivery of a Deaf awareness workshop. The provision of technology and people with lived experience meant the learning experience was not only Deaf-led, but authentic, so students became more aware of the needs of Deaf people, and were consequently more motivated to provide better care and support. |
| 7. Hoang, L., LaHousse, S. F., Nakaji, M. C., & Sadler, G. R. (2011). Assessing deaf cultural competency of physicians and medical students. *Journal of Cancer Education*, *26*(1), 175–182. USA | Survey–comparative study students who attended Deaf Community Training (DCT) program or not, included ASL classes & residential summer school | 780 medical students who attended DCT and 640 non DCT training students | To find out if medical students who attend Deaf culture training demonstrate greater knowledge of deaf culture and deaf patients than students who do not attend training | Providing healthcare providers with cultural competency training to understand that deaf communities are a linguistic and socio-cultural group will help clinicians respond more effectively to diverse communities. |

(*Continued*)

**Table 1.** (Continued)

| Study, year, location | Study design | Sample population | Study aim | Main findings |
|---|---|---|---|---|
| 8. Kruse, J., Zimmermann, A., Fuchs, M., & Rotzoll, D. (2021). Deaf awareness workshop for medical students–an evaluation. *GMS Journal for Medical Education*, 38(7). Germany | Pre and post workshop survey | 95 medical students (online workshop held on three occasions) | To determine the effect of deaf awareness training on medical students | Students reported feeling substantially more confident working with deaf people after engaging in the online deaf awareness programme. Students reported finding the deaf awareness workshop particularly helpful from a personal and from a professional point of view. |
| 9. Kung, M. S., Lozano, A., Covas, V. J., Rivera-González, L., Hernández-Blanco, Y. Y., Diaz-Algorri, Y., & Chinapen, S. (2021). Assessing Medical Students' Knowledge of the Deaf Culture and Community in Puerto Rico: a descriptive study. *Journal of medical education and curricular development*, 8, 2382120521992326. Puerto Rico | Survey testing awareness, exposure and knowledge | One student cohort at a school of medicine (n = 158 participated) | To evaluate future physician's knowledge about Deaf culture i | Overall percentage of correct answers was 39%, with knowledge limited in all groups, but some with knowledge increasing as medical students increase in experience through their course. Most frequently listed problem listed by respondents that Deaf patients may experience in hospital was fire alarm. |
| 10. Lapinski, J., Colonna, C., Sexton, P., & Richard, M. (2015). American sign language and deaf culture competency of osteopathic medical students. *American annals of the deaf*, 160(1), 36–47. USA | Cross-sectional study with pre and post test scores and evaluation | 29 students attended workshop | To examine effects of a Deaf culture workshop on Osteopathic student physicians' confidence and knowledge of working with patients using ASL | Students reported increased levels of confidence in interactions with Deaf people. 81% respondents reported the workshop as excellent, particularly enjoying the small group activities and opportunity to practice. |
| 11. Lock E. A workshop for medical students on deafness and hearing impairments. Academic Medicine. 2003 Dec 1;78(12):1229–34. Canada | Three-hour Deaf awareness workshop evaluation form | First and second year medical students | To increase awareness among physicians of the need for improved medical education on deafness and hearing impairments through a Deaf awareness workshop | Workshop evaluations suggested students found the workshop both positive and educational. Most students reported that they had not felt well informed on these subjects before the workshop, and all students stated that this type of workshop should be included in their curriculum. |
| 12. McGlade, K., Saunders, E., Thomson, C., & Woodside, J. V. (2013). Deaf awareness training in medical schools. *Medical teacher*, 35(9), 789–790. UK and Ireland | Survey | 38 medical schools in UK and Ireland (n = 38) | To examine Deaf awareness provision in medical schools in UK and Ireland | Medical schools completed survey (n = 23). 7/23 medical schools did not provide any Deaf awareness training. Of the 16 medical schools who provided training, only 8 made it compulsory. 6 provided a formal qualification in Sign Language or deaf awareness. Time spent training varied from 1–2 hours to six weeks. 13/16 involved a deaf tutor in teaching delivery. |

(*Continued*)

**Table 1.** (Continued)

| Study, year, location | Study design | Sample population | Study aim | Main findings |
|---|---|---|---|---|
| 13. Mathews, J. L., Parkhill, A. L., Schlehofer, D. A., Starr, M. J., & Barnett, S. (2011). Role-reversal exercise with deaf strong hospital to teach communication competency & cultural awareness. *American Journal of Pharmaceutical Education*, 75(3) USA | Survey | First year pharmacy students | To assess student learning of a role-reversal exercise of awareness of communication barriers with Deaf people in healthcare settings | 97% students who participated agreed or strongly agreed the experience would likely impact on their attitudes and behaviour in future interactions with patients |
| 14. O'Neill, B., Gill, E., & Brown, P. (2005). Deaf awareness and sign language: an innovative special study module. *Medical Education*, 39(5), 519–520. UK | Evaluation survey | Four groups of medical students studying short module (n = 54) | To evaluate special study module about Deaf awareness and the use of British Sign Language | 52 students completed evaluation forms, 98% students reported the sign language component as manageable. 19% students wanted more medical vocabulary in the module. Students indicated they were satisfied with being able to communicate with Deaf patients and of the opportunity to explore Deaf culture. Undergraduate medical education has a need for Deaf awareness training |
| 15. Thew D, Smith SR, Chang C, Starr M. The deaf strong hospital program: a model of diversity and inclusion training for first-year medical students. Academic Medicine. 2012 Nov 1;87 (11):1496–500. USA | Short-term and long-term post-program evaluations | Over 100 first-year medical students | To expose medical students to the Deaf Strong Hospital program to communication, linguistic, and cultural issues that are relevant to providing effective patient care and to establishing multicultural sensitivity | Since 2006, more than 90% of the students "strongly agree" or "agree" that participating in the DSH program helped them to realize the importance of the cultural, linguistic, and communication issues in delivering health care to patients from different cultures. In 2012, past participants were contacted, most respondents (37/38; 97%) recalled participating and felt that it was a valuable experience. |

Findings are reported under four deductive themes: i) provision of Deaf awareness training ii) Deaf awareness: reflections iii) Deaf awareness: examining knowledge and iv) Deaf awareness: exploring confidence and communication.

## i) Provision of Deaf awareness training

One article examined the provision of Deaf awareness training across medical schools in the UK and Ireland and was the only paper retrieved [43] to survey education providers and to ask about Deaf awareness provision. Medical schools in the UK (n = 38) were asked to complete a survey as to whether they included Deaf and hard of hearing awareness training in their curriculum, with 23 respondents [43]. 7/23 medical schools reported they did not provide any Deaf and hard of hearing awareness training, and of the 16 medical schools who said they provided training, 8 made it compulsory. 6/16 provided a formal qualification in Sign Language or Deaf and hard of hearing awareness. Time spent training varied from 1–2 hours to six weeks, and 13/16 involved a Deaf and hard

of hearing tutor in teaching delivery [43]. No other papers have been retrieved that have enquired about provider provision of Deaf awareness training for health professional programs.

### ii) Deaf awareness: Reflections

As seen in Table 1, among the 15 studies reporting Deaf awareness training, five studies delivered a workshop and undertook either a post training reflection or evaluation to enquire about participants' experiences [3,44–47] (there was no pre-testing or baseline knowledge enquiry for these studies), and all involved training delivery with members of local Deaf and hard of hearing communities. Of these evaluation studies, one involved pharmacy students [47], one further paper discussed Deaf awareness introduced across ten health professional disciplines [3], and the remaining three were conducted solely with medical students [44–46].

One study [3], led as a medical student project, was run as a collaboration with students and hard of hearing people, with general evaluations very positive, and one of the ten professional student groups who attended (occupational therapy students) completing reflective journals post workshop. The authors, who reported their interest in the logistics of delivery with patients and community partners, strongly recommend delivery of Deaf awareness training in using Deaf people as mentors to students, initially as a panel, which was replaced for subsequent sessions with mentors and students in small groups for more informal interactions.

Another study to use reflections to understand students' (pharmacy) experience of a Deaf awareness session [47], engaged participants in a different learning style with members from a nearby centre for Deaf and hard of hearing people and participation in a role-reversal exercise as students 'became' Deaf patients. Members of the Deaf and hard of hearing community wrote scenarios for student learning, and prior to the exercise students had basic lessons in American Sign Language (ASL) and reading materials about Deaf and hard of hearing culture. Students then experienced the patient perspective and different parts of a mock hospital experience as they communicated symptoms without using their voices and moved through processes of asking for interpreters, consenting to treatment, and giving symptom information. The session included debriefing, reflection on the experience and students learned the frustrating experiences in healthcare that Deaf and hard of hearing people experience [47]. 65 pharmacy students agreed the experience would positively impact their attitudes and future behaviour towards Deaf and hard of hearing patients [47]. As part of course requirements students wrote two-page reflections on the experience. In terms of feasibility the authors [47] note the nearness of the centre for Deaf people being close by helped. Authors note a small number of students were involved with requirement for heavy resource, for example 12 faculty members were involved [47].

Three further studies, that included an evaluation only type design, were focused solely on medical students [44–46]. The first of these also involved a role reversal experience for students, as well as involvement from 40 local Deaf individuals [45]. Medical students in their new 'Deaf' role interacted across four stations/types of clinical setting and were given instruction cards and waited for their 'appointment turn' as a Deaf receptionist finger-spelled their names. The program evaluated positively and at a later time point over 12 months later, 97% recalled participating and reported finding it a valuable experience [45].

The final two evaluation only studies [44,46] involved a one-off Deaf awareness workshop and both studies involved participation from Deaf community trainers, with evaluations showing that students had highly rated the activities. Time lengths of the Deaf awareness workshops varied depending on content, from one three-hour workshop run in the evening [46] to a 72-hour activity which included a short series of workshops for learning British Sign Language [44]. In response to initial positive evaluations from medical students on a Sensory awareness Day, a special study module was developed including a short Sign language course taught by a

Deaf and hard of hearing tutor and self-directed material to gain insight into Deaf and hard of hearing awareness [44]. The course included a written report, British Sign Language (BSL) tutorials and classes, a BSL objective structured clinical examination (OSCE) assessed by a certified BSL examiner, all totalling 72 hours of study activity [44]. To date 54 medical students have undertaken the course, and out of 52 completed evaluations 98% students found the sign language manageable and the content appropriate for clinical practice, although 19% would have liked more medical vocabulary [44].

### iii) Deaf awareness: Examining knowledge

Four papers reported in this section focused on examining student knowledge after a Deaf awareness session with two studies involving a control group [48,49], and two studies seeking knowledge about students existing knowledge without participation in any Deaf awareness program [50,51].

First, two studies of medical students that sought to discover if students who attend Deaf and hard of hearing culture training demonstrated greater knowledge of Deaf and hard of hearing culture and Deaf and hard of hearing patients than medical students not given a Deaf and hard of hearing awareness educational opportunity (control group), one UK study [48] and one from USA [49]. A significant difference was noted on survey questionnaires to measure attitudes to and knowledge of Deafness in those taking an optional Deaf and hard of hearing awareness course (n = 29) and control group, who could perceive patient mannerisms as rude [48]. Students reported that without encountering Deaf and hard of hearing people it may be difficult to understand the issues they face [48]. For the USA study, students were asked to list up to five problems they thought a Deaf and hard of hearing person might experience on hospitalisation, with students who had attended Deaf and hard of hearing cultural training showing awareness about understanding terms and medical language as the number one difficulty, but also acknowledging awareness about maltreatment and mistreatment being a possibility, which others in control group did not show awareness about [49].

For the two studies that sought to know students' existing knowledge about Deaf awareness without participation in a Deaf awareness program, surveys were undertaken with nursing students in the USA [50] and with medical students in Puerto Rico [51]. For nursing students [50], the survey included multiple choice questions then true/false statements [50]. Out of 131 respondents [50], 18 had taken an entry level sign language course previously. Only 17% (n = 22) answered more than half the questions correctly indicating that overall there were low levels of Deaf awareness across the cohort and low Deaf cultural competence. For the medical students' study [51] (n = 158) were asked about their knowledge of Deaf culture and community in Puerto Rico, without any intervention [51], 21% of respondents had attended a sign language class, and generally students in more senior years reported more likelihood of working with a Deaf or hard of hearing patient and showed an increased understanding of Deaf culture in comparison to junior students. Studies that indicate low baseline knowledge about a particular patient group without preparation are to be expected, but also highlight the need to increase Deaf awareness in those student populations.

### iv) Deaf awareness: Exploring confidence and communication

In this section of the scoping review findings, we report on five studies that involved a pre and post test for student groups before and after their participation in a Deaf awareness program [4,5,52–54].

The first example of a study in this review that involved USA pharmacy students [52], involved them embarking on a co-curricular course that consisted of four 90-minute sessions

including Deaf and hard of hearing cultural competence and sign language words and phrases [52], with students who completed the course reporting significantly improved knowledge and feelings of confidence in relation to communicating with people who are Deaf and hard of hearing [52]. Initially the six-hour course had a cost of $50, reduced to $12 for each student by university sponsorship. As an external agency provided and co-ordinated the courses, it is noted that the workload was not additional for course staff.

Another university applied Deaf culture to an anatomy session

With US Medical students [5], while they studied the ear and hearing [5]. Deaf and hard of hearing panellists attended this 90-minute session and discussed their healthcare experiences, additionally a further 90-minute session on Deaf and hard of hearing culture was provided, with students given pre and post session questions. Students gave positive feedback about the cultural competencies relating directly to the anatomy and neuroscience session, with students recognising their previous low knowledge levels in relation to Deaf and hard of hearing communities

Medical students in Germany [4] were invited to attend an online workshop held on three consecutive occasions, and to engage in pre and post evaluations (n = 95) [47]. 65.3% of students had not been in contact with a Deaf and hard of hearing or person before. Students reported feeling substantially more confident working with Deaf and hard of hearing people after engaging in the online Deaf and hard of hearing awareness program. Students reported finding the Deaf awareness workshop particularly helpful from a personal and from a professional point of view. The workshop was elective and the only Deaf awareness intervention that was delivered online, out of the 15 papers found in this scoping review.

Similarly, osteopathic medical students in the USA [53] participated in a pre-test, a four-hour workshop, then post-test study two weeks later with significantly improved scores at post-test following workshop attendance [53]. Students reported the contact with Deaf and hard of hearing people as part of the workshop to be the most beneficial aspect of learning, and also commended the opportunities to practice their newly learned skills.

The final study included participating nursing students in the USA using newly acquired Deaf awareness knowledge to 'assess' a 'deaf patient' after a Deaf awareness lecture [54]. This study involved students interacting directly with Deaf and hard of hearing people acting as standardised patients [54]. On starting an initial history taking exercise students were unaware patients would be Deaf and hard of hearing, mirroring real-life practice situations. Initially several students were reported to have turned away, preventing lip-reading, or left the room without saying they were going in search of interpreters. Students then received further input about communicating with Deaf communities and several positive changes were noted in the second exercise. This study [54] is another good example of how local Deaf and hard of hearing communities can be directly involved in providing students with a meaningful learning experience, which Deaf and hard of hearing participating tutors reported benefits in contributing to nurse education and improving care for others [54].

Overall papers retrieved in this scoping review suggest that health professional students who have the opportunity to engage in Deaf and hard of hearing awareness education courses during their undergraduate training find it beneficial as an opportunity to increase their knowledge about Deaf and hard of hearing people, as well as increasing their confidence and competence when communicating with Deaf and hard of hearing patients.

## Discussion

This scoping review describes the extent and characterises existing research on Deaf awareness training in health professional programs. We found that there is significant variability in how Deaf awareness training and programs exist for health professional students as well

as how the learning may be assessed and examined. Generally, health professional training does not include significant content about learning how to communicate with Deaf and hard of hearing people and few opportunities to develop Deaf and hard of hearing cultural competencies. The lack of content regarding the care of Deaf and hard of hearing people during education of all health professional students may be one of the explanations for the difficulty of interaction between professionals and the dissatisfaction Deaf and hard of hearing users of health services experience [55,56]. Evidence retrieved usually involved small samples, and providers were often supported by external agencies in terms of delivery of Deaf awareness training initiatives.

Several of the retrieved studies reported on one-off interventions with small participant numbers, some of which required heavy resource from either education faculties, local Deaf and hard of hearing centres or both [47,50,52]. Whilst direct involvement from Deaf and hard of hearing communities is admirable and probably the best experience for student learning, it may not be feasible for health professional programs to aspire to such learning opportunities due to high numbers of students. Providing the opportunities to a select few is not in the spirit of equity, and Deaf awareness knowledge and cultural competence surely need to be known to all undertaking a health professional program. Education providers with large student populations simply cannot over-burden local Deaf and hard of hearing communities to come on-site and provide teaching and practice opportunities, and the logistics of organising this for large cohorts are challenging, with providers aware of competing topics, and limited program time. One solution would be for the development of Deaf awareness eLearning packages that have been Deaf-led and include the development of knowledge about types of Deafness, best ways to communicate, what to avoid, as well as promoting positivity around Deaf and hard of hearing population, so that Deaf culture is not only learned about, but embraced.

In terms of approaches, it is unsurprising that Deaf awareness interventions increased student knowledge and cultural competence about working with Deaf and hard of hearing people. Collecting pre and post knowledge information would certainly demonstrate a more robust approach and supply feedback about the impact of interventions, as well as the opportunity for students to apply their Deaf awareness as evidence-informed practice [57].

A solution by some providers in terms of navigating competing timetable demands is to provide Deaf awareness as optional [48] resulting in probably the most motivated students attending, and again resulting in the student majority not having the opportunity for Deaf and hard of hearing awareness skill and knowledge development. Yet health professional comfort levels at communicating with Deaf and hard of hearing patients increase when they have more contact with Deaf and hard of hearing patients [58].

As with most skills workshops, and several of the studies in this review included a student opportunity to learn basic signed language, it is acknowledged that unless learners have the opportunity to regularly practice a skill it may soon be lost [58], so a thorough approach with regularity and informal practice time would be essential for success.

A scarcity of evidence was found from allied health professional programs regarding Deaf awareness content. This is notable in terms of audiology student programs, although anecdotally many claim to include a session on the topic. Regarding qualified audiologists, two studies examined audiologists' current cultural competency [59] and the need for audiologists to have clinically relevant sign language [60]. There is a need for audiologists to increase their knowledge of Deaf awareness and proficiency in sign language starting during their professional training is clear, as well as their knowledge regarding how to work with sign language interpreters [59], which applies to all healthcare professionals. Similarly, others who work with patient groups, such as genetic counselling graduates [61], with over a quarter reporting no

Deaf and hard of hearing awareness training and 51% reported limited training of just 1–2 hours during their initial training programs.

There may be certain professional groups who are viewed as more likely to encounter contact directly with a Deaf and hard of hearing person. For example in a study of emergency medical practitioners [62], out of 148 respondents, 109 reported having responded to an emergency call from a Deaf and hard of hearing person. In the same study, participants who attended training said it expanded their knowledge of Deaf and hard of hearing culture; and at 3 months all respondents reported the training to still be helpful and clinically relevant.

Any facilitators of Deaf awareness programs need to ensure accuracy in terms of context and relevant country/regional Sign language. Assumptions are often made, for example a study about Deaf awareness training for support staff with people with intellectual disabilities [63] talked about using signs but people are not always aware that sign for communication support differ considerably a recognised Signed language. For example, Makaton is not a recognised language but is a communication tool [64].

Health professional students themselves have noted that workshops similar to Deaf awareness would help considerably in increasing their knowledge and skills of how best to communication and work with under-served populations [51]. Despite moves to progress accessible standards in health services, we continue to know that populations continue to have poor experiences in healthcare which mostly relate to the limited knowledge and preparedness of those working in such professions.

### Recommendations

Involving Deaf and hard of hearing individuals in the planning and delivery of activities at the outset will ensure content is accurate, relevant and may provide the opportunity for immediate feedback if practical exercises are included. Deaf awareness training during health professional training programs can serve as a timely introduction to the topic and ensure students thoughtfully evaluate their approach to communication and engagement with all individuals. At the very minimum all health professional programs need to provide basic Deaf awareness information that can be accessed on their student learning platforms, along with information about local Deaf communities and Sign Language training providers.

### Strengths and limitations of this study

- Using the guidelines of the PRISMA-ScR (Preferred Reporting Items for Systematic Reviews and Meta-Analyses extension for Scoping Reviews), this study provides a detailed view of the evidence of Deaf and hard of hearing awareness content that feature in health professional student programs

- Literature from four electronic databases and registers were screened to comprehensively source and describe the literature.

- Only published peer-reviewed research articles in English were included (although initial searches for papers in other languages did take place with none located)

- Despite a systematic approach, there is a risk that further evidence may have been overlooked.

### Conclusion

As Deaf and hard of hearing communities frequently report negative experiences in healthcare largely due to a lack of Deaf awareness knowledge in staff, it is important to understand more

about the delivery of Deaf awareness programmes available to health workers. This scoping review outlined the available evidence regarding health professional programs that include Deaf awareness content aimed to increase students' knowledge skills and Deaf cultural competencies as they move forwards in their careers. There is a lack of rigorous research in this field, although there is emerging evidence of benefits and increased Deaf awareness knowledge for student populations. All development of Deaf awareness education needs full involvement from Deaf and hard of hearing communities to ensure relevance and success. Programme regulators and providers have an important role here in reviewing program content to ensure disadvantaged communities do not remain under-served. There is potential to ensure that students emerge from health professional education with good knowledge about how to work with Deaf and hard of hearing patients.

Our review offers a starting point to educators and health and care providers to consider potential benefits to both health professional students and staff about increasing knowledge, confidence and competence about Deaf awareness, as well as modes of delivery. Further research on the acceptability of, and implementation of Deaf awareness programs on health professional students is needed. Knowledge gaps exist around the type of Deaf awareness programs, how such training might be accessed, length of course, content and device delivery. Knowing ultimately what communication approaches impact positively on Deaf people in healthcare services is the ultimate goal.

## Supporting information

**S1 Checklist. PRISMA checklist.**
(DOCX)

**S1 Text. Cinahl search.**
(DOCX)

## Acknowledgments

The authors thank the project steering group, particularly Deaf and hard of hearing communities in Wales for their support and interest with the project.

## Author Contributions

**Conceptualization:** Julia Terry, Rhian Meara.

**Data curation:** Julia Terry.

**Formal analysis:** Julia Terry.

**Funding acquisition:** Julia Terry.

**Investigation:** Julia Terry.

**Methodology:** Julia Terry, Rhian Meara.

**Project administration:** Julia Terry.

**Resources:** Julia Terry.

**Visualization:** Julia Terry.

**Writing – original draft:** Julia Terry.

**Writing – review & editing:** Julia Terry, Rhian Meara.

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
