## [Decision Letter · Decision Letter 0]

12 Mar 2024

PGPH-D-23-02572

A Scoping Review of Deaf Awareness Programs in Health Professional Education

Dear Dr. Terry,

Thank you for submitting your manuscript to PLOS Global Public Health. After careful consideration, we feel that it has merit but does not fully meet PLOS Global Public Health’s publication criteria as it currently stands. Therefore, we invite you to submit a revised version of the manuscript that addresses the points raised during the review process.

Please see the comments of one reviewer below. The reviewer seems overall positive about the contributions of the manuscript, but has pointed out a range of suggestions for strengthening the study.

Please note that we have only been able to secure a single reviewer to assess your manuscript. We are issuing a decision on your manuscript at this point to prevent further delays in the evaluation of your manuscript. Please be aware that the editor who handles your revised manuscript might find it necessary to invite additional reviewers to assess this work once the revised manuscript is submitted. However, we will aim to proceed on the basis of this single review if possible. 

We look forward to receiving your revised manuscript.

Kind regards,

Hanna Landenmark

Staff Editor

Journal Requirements:

1. We noticed you have some minor occurrence of overlapping text with the following previous publication(s), which needs to be addressed:

- https://doi.org/10.1111/tct.13304

- https://doi.org/10.1111/j.1365-2929.2005.02153.x

In your revision ensure you cite all your sources (including your own works), and quote or rephrase any duplicated text outside the methods section. Further consideration is dependent on these concerns being addressed.

2. Please send a completed 'Competing Interests' statement, including any COIs declared by your co-authors. If you have no competing interests to declare, please state "The authors have declared that no competing interests exist". 

3. "Please provide a/amend your detailed Financial Disclosure statement. This is published with the article. It must therefore be completed in full sentences and contain the exact wording you wish to be published.

4. Please provide separate figure files in .tif or .eps format and remove the embedded figures from the manuscript file.

5. Some material included in your submission may be copyrighted. According to PLOS’s copyright policy, authors who use figures or other material (e.g., graphics, clipart, maps) from another author or copyright holder must demonstrate or obtain permission to publish this material under the Creative Commons Attribution 4.0 International (CC BY 4.0) License used by PLOS journals. Please closely review the details of PLOS’s copyright requirements here: PLOS Licenses and Copyright. If you need to request permissions from a copyright holder, you may use PLOS's Copyright Content Permission form.

Potential Copyright Issues:

Figure 2: please (a) provide a direct link to the base layer of the map (i.e., the country or region border shape) and ensure this is also included in the figure legend; and (b) provide a link to the terms of use / license information for the base layer image or shapefile. We cannot publish proprietary or copyrighted maps (e.g. Google Maps, Mapquest) and the terms of use for your map base layer must be compatible with our CC-BY 4.0 license. 

6. We note that your Data Availability Statement is currently as follows: "Data is provided in the figures, table and additional files."

7. We have noticed that you have uploaded Supporting Information files, but you have not included a list of legends. Please add a full list of legends for your Supporting Information files after the references list. 

Additional Editor Comments (if provided):

Reviewers' comments:

Reviewer's Responses to Questions

**Comments to the Author**

1. Does this manuscript meet PLOS Global Public Health’s publication criteria? Is the manuscript technically sound, and do the data support the conclusions? The manuscript must describe methodologically and ethically rigorous research with conclusions that are appropriately drawn based on the data presented.

Reviewer #1: Partly

2. Has the statistical analysis been performed appropriately and rigorously?

Reviewer #1: N/A

3. Have the authors made all data underlying the findings in their manuscript fully available (please refer to the Data Availability Statement at the start of the manuscript PDF file)?

Reviewer #1: No

4. Is the manuscript presented in an intelligible fashion and written in standard English?

Reviewer #1: Yes

5. Review Comments to the Author

Reviewer #1: I would recommend a major revision – given the importance of the topic area, rather than rejection. However, lots of work is required to make this manuscript publication ready.

Overall/General comments:

• This is an under-studied area, and challenges for Deaf people accessing healthcare services are important to highlight and consider approaches for change. As such, it could be an important contribution to the debates on access, inclusion and equity.

• I commend the authors for the co-participation of a Deaf and hard of hearing steering group

• Methods: The team appear to have taken a rigorous approach to screening and reported this according to guidelines. However, it is not clear from the search strategy how health professional education was brought in, or whether this was filtered in the title/abstract review stage. List of databases accessed should be provided.

• As currently written, the manuscript is wordy and difficult to follow. I would suggest more clear paragraphs in introduction and discussion that bring together fewer themes with stronger organization.

o For example, introduction could have the following 3 paragraph themes:

What is known about provider-oriented disability awareness and access support programmes (what has worked, for whom, where)

Challenges to access for Deaf and hard of hearing patients in health and how this is exacerbated by gaps in provider awareness

How this leads to the gap in knowledge around Deaf awareness programmes for health professionals that made this study

o Similarly the discussion needs to be organized around key themes that are supported by the paper’s findings; consider including some recommendations and bringing that out separately from the discussion.

• Results: this section is organized around the type of health professional students receiving the intervention. Please go back and instead consider thematic analysis of the findings showing what is consistent or variable across the different types of programmes / students / settings

• I note that the funder is simply stated as ‘xxxx’ – please edit.

6. PLOS authors have the option to publish the peer review history of their article (what does this mean?). If published, this will include your full peer review and any attached files.

**Do you want your identity to be public for this peer review?** For information about this choice, including consent withdrawal, please see our Privacy Policy.

Reviewer #1: No

---

## [Decision Letter · Decision Letter 1]

3 Jun 2024

PGPH-D-23-02572R1

A Scoping Review of Deaf Awareness Programs in Health Professional Education

Dear Dr. Terry,

Thank you for submitting your manuscript to PLOS Global Public Health. After careful consideration, we feel that it has merit but does not fully meet PLOS Global Public Health’s publication criteria as it currently stands. Therefore, we invite you to submit a revised version of the manuscript that addresses the points raised during the review process.

Kindly address the outstanding comments raised by the reviewer.

We look forward to receiving your revised manuscript.

Kind regards,

Avanti Dey, PhD

Staff Editor

Journal Requirements:

Additional Editor Comments (if provided):

Reviewers' comments:

Reviewer's Responses to Questions

**Comments to the Author**

1. If the authors have adequately addressed your comments raised in a previous round of review and you feel that this manuscript is now acceptable for publication, you may indicate that here to bypass the “Comments to the Author” section, enter your conflict of interest statement in the “Confidential to Editor” section, and submit your "Accept" recommendation.

Reviewer #2: (No Response)

2. Does this manuscript meet PLOS Global Public Health’s publication criteria? Is the manuscript technically sound, and do the data support the conclusions? The manuscript must describe methodologically and ethically rigorous research with conclusions that are appropriately drawn based on the data presented.

Reviewer #2: Yes

3. Has the statistical analysis been performed appropriately and rigorously?

Reviewer #2: N/A

4. Have the authors made all data underlying the findings in their manuscript fully available (please refer to the Data Availability Statement at the start of the manuscript PDF file)?

Reviewer #2: (No Response)

5. Is the manuscript presented in an intelligible fashion and written in standard English?

Reviewer #2: Yes

6. Review Comments to the Author

Reviewer #2: This scoping review contributes towards important research about awareness in future medical professionals about diverse patient needs and is a great approach for designing an awareness program in Wales in the future. It explores existing Deaf awareness programs for health professional students in a very effective way. The authors use a well-established framework, and transparently outline their methods. The strengths of this manuscript include a well-defined research question, a broad search strategy with diverse resources, and clear methodology using PRISMA and PICO.

Areas for improvement include acknowledging and rationale for excluding non-English studies and the challenges due to study heterogeneity. Further, the gaps for further research could be more clearly presented.

Overall, this is a well-designed scoping review with valuable insights into Deaf awareness programs. Addressing the daily challenges Deaf and hard of hearing patients face in medical contexts.

„Many diverse groups are disadvantaged because of assumptions around health literacy that may relate to English not being a first language, ability to read and write, which in turn impacts on a person’s ability to understand healthcare and pharmacy directions and information“ - page 6, please clarify. This is an important point about assumptions around health literacy that could be made stronger.

In the end of the introduction: „address this knowledge gap“ Does this refer to the gap in research or the gap in the knowledge of the students?

7. PLOS authors have the option to publish the peer review history of their article (what does this mean?). If published, this will include your full peer review and any attached files.

**Do you want your identity to be public for this peer review?** For information about this choice, including consent withdrawal, please see our Privacy Policy.

Reviewer #2: No

---

## [Decision Letter · Decision Letter 2]

20 Jun 2024

PGPH-D-23-02572R2

A Scoping Review of Deaf Awareness Programs in Health Professional Education

Dear Dr. Terry,

Thank you for submitting your manuscript to PLOS Global Public Health. After careful consideration, we feel that it has merit but does not fully meet PLOS Global Public Health’s publication criteria as it currently stands. Therefore, we invite you to submit a revised version of the manuscript that addresses the points raised during the review process.

The reviewer has suggested some final very minor comments, please attend to these before the manuscript can be accepted.

We look forward to receiving your revised manuscript.

Kind regards,

Avanti Dey, PhD

Staff Editor

Journal Requirements:

Additional Editor Comments (if provided):

Reviewers' comments:

Reviewer's Responses to Questions

**Comments to the Author**

1. If the authors have adequately addressed your comments raised in a previous round of review and you feel that this manuscript is now acceptable for publication, you may indicate that here to bypass the “Comments to the Author” section, enter your conflict of interest statement in the “Confidential to Editor” section, and submit your "Accept" recommendation.

Reviewer #2: All comments have been addressed

2. Does this manuscript meet PLOS Global Public Health’s publication criteria? Is the manuscript technically sound, and do the data support the conclusions? The manuscript must describe methodologically and ethically rigorous research with conclusions that are appropriately drawn based on the data presented.

Reviewer #2: Yes

3. Has the statistical analysis been performed appropriately and rigorously?

Reviewer #2: N/A

4. Have the authors made all data underlying the findings in their manuscript fully available (please refer to the Data Availability Statement at the start of the manuscript PDF file)?

Reviewer #2: (No Response)

5. Is the manuscript presented in an intelligible fashion and written in standard English?

Reviewer #2: Yes

6. Review Comments to the Author

Reviewer #2: Thank you for attending to all prior comments. I strongly belief this is very important research and adds value to the research corpus. As I was reviewing your paper again, I found a few paragraphs that were not consistent in the presentation of the thought process. Concluding sentences in the end of the paragraph suddenly introduced new ideas. It becomes very clear that the authors have undergone a very detailed thought process, but this is not always reflected in the written text.

Please take some time to review the text, make it consistent, and guide the reader along your line of arguments. What is your aim and who is your target audience?

I am certain these revision can help to strengthen your findings and conclusion.

7. PLOS authors have the option to publish the peer review history of their article (what does this mean?). If published, this will include your full peer review and any attached files.

**Do you want your identity to be public for this peer review?** For information about this choice, including consent withdrawal, please see our Privacy Policy.

Reviewer #2: No

---

## [Decision Letter · Decision Letter 3]

10 Jul 2024

A Scoping Review of Deaf Awareness Programs in Health Professional Education

PGPH-D-23-02572R3

Dear Dr Terry,

We are pleased to inform you that your manuscript 'A Scoping Review of Deaf Awareness Programs in Health Professional Education' has been provisionally accepted for publication in PLOS Global Public Health.

Best regards,

Julia Robinson

Executive Editor

Reviewer Comments (if any, and for reference):

Reviewer's Responses to Questions

**Comments to the Author**

1. If the authors have adequately addressed your comments raised in a previous round of review and you feel that this manuscript is now acceptable for publication, you may indicate that here to bypass the “Comments to the Author” section, enter your conflict of interest statement in the “Confidential to Editor” section, and submit your "Accept" recommendation.

Reviewer #2: All comments have been addressed

2. Does this manuscript meet PLOS Global Public Health’s publication criteria? Is the manuscript technically sound, and do the data support the conclusions? The manuscript must describe methodologically and ethically rigorous research with conclusions that are appropriately drawn based on the data presented.

Reviewer #2: Yes

3. Has the statistical analysis been performed appropriately and rigorously?

Reviewer #2: N/A

4. Have the authors made all data underlying the findings in their manuscript fully available (please refer to the Data Availability Statement at the start of the manuscript PDF file)?

Reviewer #2: (No Response)

5. Is the manuscript presented in an intelligible fashion and written in standard English?

Reviewer #2: Yes

6. Review Comments to the Author

Reviewer #2: Thank you for taking the time to revise the manuscript. I do feel this has improved how your findings come across and conveys a more clear message to the reader.

7. PLOS authors have the option to publish the peer review history of their article (what does this mean?). If published, this will include your full peer review and any attached files.

**Do you want your identity to be public for this peer review?** For information about this choice, including consent withdrawal, please see our Privacy Policy.

Reviewer #2: No
